# Deep Neural Network Fingerprinting by Conferrable Adversarial Examples

**Nils Lukas, Yuxuan Zhang, Florian Kerschbaum**
University of Waterloo
{`nlukas, y2536zhang, florian.kerschbaum`}@uwaterloo.ca

## Abstract

In Machine Learning as a Service, a provider trains a deep neural network and gives many users access. The hosted (source) model is susceptible to model stealing attacks, where an adversary derives a *surrogate model* from API access to the source model. For post hoc detection of such attacks, the provider needs a robust method to determine whether a suspect model is a surrogate of their model. We propose a fingerprinting method for deep neural network classifiers that extracts a set of inputs from the source model so that only surrogates agree with the source model on the classification of such inputs. These inputs are a subclass of transferable adversarial examples which we call *conferrable* adversarial examples that exclusively transfer with a target label from a source model to its surrogates. We propose a new method to generate these conferrable adversarial examples. We present an extensive study on the irremovability of our fingerprint against fine-tuning, weight pruning, retraining, retraining with different architectures, three model extraction attacks from related work, transfer learning, adversarial training, and two new adaptive attacks. Our fingerprint is robust against distillation, related model extraction attacks, and even transfer learning when the attacker has no access to the model provider's dataset. Our fingerprint is the first method that reaches a ROC AUC of 1.0 in verifying surrogates, compared to a ROC AUC of 0.63 by previous fingerprints.

## 1 Introduction

Deep neural network (DNN) classifiers have become indispensable tools for addressing practically relevant problems, such as autonomous driving (Tian et al., 2018), natural language processing (Young et al., 2018) and health care predictions (Esteva et al., 2019). While a DNN provides substantial utility, training a DNN is costly because of data preparation (collection, organization, and cleaning) and computational resources required for validation of a model (Press, 2016). For this reason,

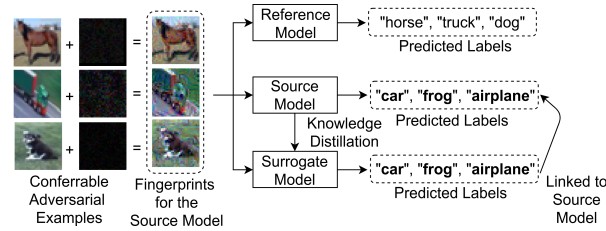

Figure 1: A set of conferrable adversarial examples used as a fingerprint to identify surrogate models.

DNNs are often provided by a single entity and consumed by many, such as in the context of Machine Learning as a Service (MLaaS). A threat to the provider is *model stealing*, in which an adversary derives a *surrogate model* from only API access to a *source model*. We refer to an independently trained model for the same task as a *reference model*.

Consider a MLaaS provider that wants to protect their service and hence restrict its redistribution, e.g., through a contractual usage agreement because trained models constitute their intellectual property. A threat to the model provider is an attacker who derives surrogate models and publicly deploys them. Since access to the source model has to be provided, users cannot be prevented from deriving surrogate models. Krishna et al. (2019) have shown that model stealing is (i) effective, because even

high-fidelity surrogates of large models like BERT can be stolen, and (ii) efficient, because surrogate models can be derived for a fraction of the costs with limited access to domain data.

This paper proposes a DNN fingerprinting method to predict whether a model is a (stolen) surrogate or a (benign) reference model relative to a source model. DNN fingerprinting is a new area of research that extracts a persistent, identifying code (fingerprint) from an already trained model. Model stealing can be categorized into *model modification*, such as weight pruning (Zhu & Gupta, 2017), or *model extraction* that uses some form of knowledge distillation (Hinton et al., 2015) to derive a surrogate from scratch. Claimed security properties of existing defenses ((Adi et al., 2018; Zhang et al., 2018)), have been broken by model extraction attacks (Shafieinejad et al., 2019). Our fingerprinting method is the first passive defense that is specifically designed towards withstanding model extraction attacks, which extends to robustness against model modification attacks.

Our research provides new insight into the transferability of adversarial examples. In this paper, we hypothesize that there exists a subclass of targeted, transferable, adversarial examples that transfer exclusively to surrogate models, but not to reference models. We call this subclass *conferrable*. Any conferrable example found in the source model should have the same misclassification in a surrogate model, but a different one in reference models. We propose a metric to measure conferrability and an ensemble adversarial attack that optimizes this new metric. We generate conferrable examples as the source model's fingerprint.

Retrained CIFAR-10 surrogate models can be verified with a perfect ROC AUC of $1.0$ using our fingerprint, compared to an ROC AUC of $0.63$ for related work (Cao et al., 2019). While our fingerprinting scheme is robust to almost all derivation and extraction attacks, we show that some adapted attacks may remove our fingerprint. Specifically, our fingerprint is not robust to transfer learning when the attacker has access to a model pre-trained on ImageNet32 and access to CIFAR-10 domain data. Our fingerprint is also not robust against adversarial training (Madry et al., 2017) from scratch. Adversarial training is an adapted model extraction attack specifically designed to limit the transferability of adversarial examples. We hypothesize that incorporating adversarial training into the generation process of conferrable adversarial examples may lead to higher robustness against this attack.

## 2 RELATED WORK

In black-box adversarial attacks (Papernot et al., 2017; Tramèr et al., 2016; Madry et al., 2017), access to the target model is limited, meaning that the target architecture is unknown and computing gradients directly is not possible. Transfer-based adversarial attacks (Papernot et al., 2016; 2017) exploit the ability of an adversarial example to *transfer* across models with similar decision boundaries. *Targeted* transferability additionally specifies the target class of the adversarial example. Our proposed adversarial attack is a targeted, transfer-based attack with white-box access to a source model (that should be defended), but black-box access to the stolen model derived by the attacker.

Liu et al. (2016) and Tramèr et al. (2017a) show that (targeted) transferability can be boosted by optimizing over an ensemble of models. Our attack also optimizes over an ensemble of models to maximize transferability to stolen surrogate models, while minimizing transferability to independently trained models, called reference models. We refer to this special subclass of targeted transferability as *conferrable*. Tramèr et al. (2017a) empirically study transferability and find that transferable adversarial examples are located in the intersection of high-dimensional "adversarial subspaces" across models. We further their studies and show that (i) stolen models apprehend adversarial vulnerabilities from the source model and (ii) parts of these subspaces, in which conferrable examples are located, can be used in practice to predict whether a model has been stolen.

Watermarking of DNNs is a related method to DNN fingerprinting where an identifying code is embedded into a DNN, thereby potentially impacting the model's utility. Uchida et al. (2017) embed a secret message into the source model's weight parameters, but require white-box access to the model's parameters for the watermark verification. Adi et al. (2018) and Zhang et al. (2018) propose backdooring the source model on a set of unrelated or slightly modified images. Their approaches allow black-box verification that only requires API access to the watermarked model. Frontier-Stitching (Merrer et al., 2017) and BlackMarks (Dong et al., 2018) use (targeted) adversarial examples as watermarks. These watermarks have been evaluated only against model modification

attacks, but not against model extraction attacks that train a surrogate model from scratch. At least two of these watermarking schemes (Adi et al., 2018; Zhang et al., 2018) are not robust to model extraction attacks (Shafieinejad et al., 2019). Cao et al. (2019) recently proposed a fingerprinting method with adversarial examples close to the source model's decision boundary. We show that their fingerprint does not withstand retraining as a model extraction attack and propose a fingerprint with improved robustness to model extraction attacks.

## 3 DNN FINGERPRINTING

**Threat Model.** The attacker's goal is to derive a surrogate model from access to the defender's source model that (i) has a similar performance (measured by test accuracy) and (ii) is not verified as a surrogate of the source model by the defender. We protect against an informed attacker that can have (i) white-box access to the source model, (ii) unbounded computational capacity and (iii) access to domain data from the same distribution. A more informed attacker can drop information and invoke all attacks of a less informed attacker. Robustness against a more informed attacker implies robustness against a less informed attacker. Our attacker is limited in their access to ground-truth labeled data, otherwise they could train their own model and there would be no need to steal a model. In our evaluation, we experiment with attackers that have up to 80% of ground-truth labels for CIFAR-10 (Krizhevsky et al.). A practical explanation for the limited accessibility of ground truth labels could be that attackers do not have access to a reliable oracle (e.g., for medical applications), or acquiring labels may be associated with high expenses (e.g., Amazon Mechanical Turk[1]).

The defender's goal is to identify stolen surrogate models that are remotely deployed by the attacker. Their capabilities are (i) white-box access to the source model, (ii) black-box access to the target model deployed by the attacker and (iii) a limited set of $n$ queries to verify a remotely deployed surrogate model. Black-box access to the suspect model excludes knowledge of the suspect model's architecture or attack used to distill the model. The defender also does not have knowledge of, nor control over the attacker's dataset used to steal the model.

**Fingerprinting Definitions.** A fingerprinting method for DNNs consists of two algorithms: (i) A fingerprint *generation* algorithm that generates a secret fingerprint $\mathcal{F} \subseteq \mathcal{X}^n$ of size $n$, and a fingerprint verification key $\mathcal{F}_y \subseteq \mathcal{Y}^n$; (ii) A fingerprint *verification* algorithm that verifies surrogates of the source model. These algorithms can be summarized as follows.

- **Generate**$(M, D)$: Given white-box access to a source model $M$ and training data $D \in \mathcal{D}$. Outputs a fingerprint $\mathcal{F}$ and the verification keys $\mathcal{F}_y = \{M(x) | x \in \mathcal{F}\}$.

- **Verify**$(\hat{M}(\mathcal{F}), \mathcal{F}_y)$: Given black-box access to a suspect model $\hat{M}$, a fingerprint $\mathcal{F}$ and a verification key $\mathcal{F}_y$. Outputs 1 if $\hat{M}$ is verified by the fingerprint and 0 otherwise.

The verification algorithm computes an error rate between the outputs of the source and target model on the fingerprint. We empirically measure the error rate separately for surrogate and reference models, which allows to choose a decision threshold $\rho \in [0, 1]$. If the error-rate of a target model exceeds $1 - \rho$, the verification predicts the target model to be a reference model, otherwise the prediction is a surrogate model. We define that a fingerprint must be *irremovable*, i.e., surrogate and reference models are correctly verified, despite an attacker's removal attempt. It must also be *non-evasive*, meaning that an attacker cannot evade black-box verification by detecting fingerprint queries. We refer to Appendix A.3 for security games of DNN fingerprinting.

## 4 CONFERRABLE ADVERSARIAL EXAMPLES

**Motivating Conferrability** Conferrability is a new property for adversarial examples, in which targeted transferability occurs only from a source model to its surrogates, but not to independently trained reference models. Intuitively, surrogate models are expected to be more similar to the source model than any reference model, but quantifying this similarity is non-trivial. Conferrable examples are an attempt to quantify this similarity by shared adversarial vulnerabilities of the source model

---

[1]https://www.mturk.com/

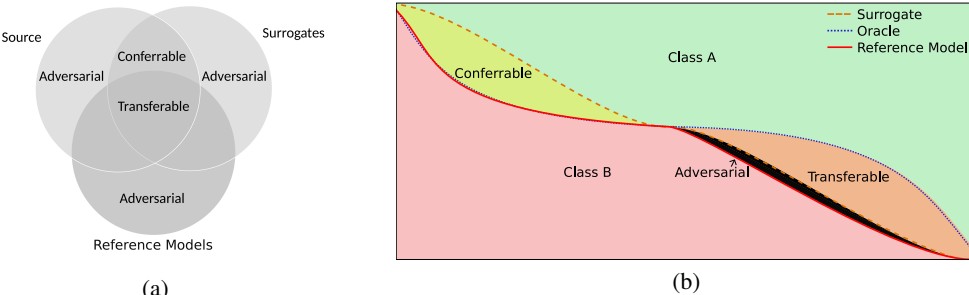

Figure 2: (a) A summary of the relationship between transferability and conferrability as intersections between the set of all adversarial examples per model type. (b) A representation of transferable and conferrable examples in the decision space, relative to the ground-truth provided by an oracle.

and its surrogates. Surrogate models differ from reference models in the objective function which they optimize. Reference models maximize fidelity to the ground truth labels, whereas surrogate models maximize fidelity to the source model's labels, which do not always coincide.

Fig. 2a shows the relation between targeted adversarial, transferable, and conferrable examples for the source model, its surrogates, and all reference models. An example is transferable if it is adversarial to any model. It is conferrable when it is adversarial only to surrogate and source models. In that sense, conferrable adversarial examples are a subclass of targeted transferable adversarial examples. Fig. 2b shows the relation between transferable and conferrable adversarial examples in the decision space, simplified for illustrative purposes to binary classification. Transferable adversarial examples occur in those adversarial subspaces where the decision boundary of surrogate and reference models coincide (Tramèr et al., 2017b). Conferrable adversarial examples occur in those adversarial subspaces where the decision boundary of surrogate and reference models differ.

Targeted transferability for a class $t$ and a set of models $\mathcal{M}$ can be computed as follows.

$$\text{Transfer}(\mathcal{M}, x; t) = \Pr_{M \in \mathcal{M}} [M(x) = t] \tag{1}$$

**Objective** We want to find adversarial examples that maximize the output activation difference between surrogate models $\mathcal{S}$ and reference models $\mathcal{R}$. This difference is quantified by our *conferrability* score, which measures the example's transferability to surrogate and reference models.

$$\text{Confer}(\mathcal{S}, \mathcal{R}, x; t) = \text{Transfer}(\mathcal{S}, x; t)(1 - \text{Transfer}(\mathcal{R}, x; t)) \tag{2}$$

The central challenge for optimizing conferrability directly is that we require access to a differentiable function *Transfer* that estimates an example's transferability score. To the best of our knowledge, the only known method to evaluate transferability is to train a representative set of DNNs and estimating the example's transferability itself. We use this method to evaluate transferability. In the case of conferrable examples, we evaluate transferability on a set of surrogate and reference models trained locally by the defender and use Equation 2 to obtain the conferrability score. Our hypothesis is that conferrability generalizes, i.e., examples that are conferrable to a representative set of surrogate and reference DNNs are also conferrable to other, unseen DNNs.

**Conferrable Ensemble Method.** In this section, we describe our adversarial attack that generates conferrable adversarial examples, called the *Conferrable Ensemble Method* (CEM). CEM operates on the source model $M$, a set of its surrogates $\mathcal{S}_M$, and a set of reference models $\mathcal{R}$. The attack constructs an ensemble model $M_E$ with a single shared input that outputs conferrability scores for all output classes $y \in \mathcal{Y}$. CEM generates highly conferrable adversarial examples by finding a perturbation $\delta$ so that $x' = x_0 + \delta$ maximizes the output of the ensemble model for a target class $t$.

We now present the construction of the ensemble model $M_E$. The ensemble model produces two intermediate outputs for an input $x \in \mathcal{X}$, representing the average predictions of all surrogate and

all reference models on $x$. We use Dropout (Srivastava et al., 2014) with a drop ratio of $d = 0.3$.

$$\text{Surr}(\mathcal{S}_M, x) = \frac{1}{|\mathcal{S}_M|} \sum_{S \in \mathcal{S}_M} \text{Dropout}(S(x); d) \tag{3}$$

$$\text{Ref}(\mathcal{R}, x) = \frac{1}{|\mathcal{R}|} \sum_{R \in \mathcal{R}} \text{Dropout}(R(x); d) \tag{4}$$

The output of the ensemble model is the conferrability score of the input for each class. The Softmax activation function is denoted by $\sigma$. We refer to Appendix A.4 for details about the optimization.

$$M_E(x; \mathcal{S}_M, \mathcal{R}) = \sigma(\text{Surr}(\mathcal{S}_M, x)(1 - \text{Ref}(\mathcal{R}, x))) \tag{5}$$

The loss function consists of three summands and is computed over the benign, initial example $x_0$, and the example at an intermediate iteration step $x' = x_0 + \delta$. We denote the cross-entropy loss by $H(\cdot, \cdot)$. The first summand function maximizes the output of the ensemble model for some target class $t$. The second summand maximizes the categorical cross-entropy between the current and initial prediction for the source model. The third summand minimizes the categorical cross-entropy between the source model's prediction and the predictions of its surrogates. The total loss $\mathcal{L}$ is the weighted sum over all individual losses.

$$\mathcal{L}(x_0, x') = \alpha H(1, \max_t[M_E(x')_t]) - \beta H(M(x_0), M(x')) + \gamma H(M(x'), \text{Surr}(\mathcal{S}_M, x')) \tag{6}$$

In all our experiments, we use weights $\alpha = \beta = \gamma = 1$ and refer to Appendix A.6 for an empirical sensitivity analysis of the hyperparameters. We address the box constraint, i.e., $||\delta|| \leq \epsilon$, similarly to PGD (Madry et al., 2017) by clipping intermediate outputs around the $\epsilon$ ball of the original input $x_0$. For the optimization of an input with respect to the loss we use the Adam optimizer (Kingma & Ba, 2014). Note that we use an untargeted attack to generate targeted adversarial examples. Targeted attacks are harder to optimize than their untargeted counterparts (Wu & Fu, 2019). We assign the source model's predicted label for the generated adversarial example as the target label and ensure that target classes are balanced in the fingerprint verification key.

**Fingerprinting Algorithms.** We now describe our fingerprinting generation and verification algorithms. For the generation, the defender locally trains a set of $c_1$ surrogate and $c_2$ reference models ($c_1 = c_2 = 18$) on their training data prior to executing CEM. Surrogate models are trained on data labeled by the source model, whereas reference models are trained on ground-truth labels. The defender composes the ensemble model $M_E$ as described in the previous paragraph and optimizes for a perturbation $\delta$ given an input $x_0$, so that $\mathcal{L}(x_0, x_0 + \delta)$ is minimized. The optimization returns a set of adversarial examples that are filtered by their conferrability scores on the locally trained models. If their conferrability score (see Equation 2) exceeds a minimum threshold ($\tau \geq 0.95$), they are added to the fingerprint. For the fingerprint verification, we compute the error rate between the source model's prediction on the fingerprint and a target model's predictions. If the error rate is greater than $1 - \rho$, which we refer to as the *decision threshold*, the target model is predicted to be a reference model and a surrogate model otherwise.

## 5 EXPERIMENTAL SETUP

**Experimental Procedure.** We evaluate the irremovability and non-evasiveness of our fingerprint. We show that our proposed adversarial attack CEM improves upon the mean conferrability scores compared to other adversarial attacks, such as FGM (Goodfellow et al., 2014), BIM (Kurakin et al., 2016), PGD (Madry et al., 2017), and CW-$L_\infty$ (Carlini & Wagner, 2017b). We demonstrate the non-evasiveness of our fingerprint by evaluating against the evasion algorithm proposed by Hitaj et al. (2019). Our study on irremovability is the most extensive study conducted on DNN fingerprints or DNN watermarks compared to related work (Cao et al., 2019; Uchida et al., 2017; Zhang et al., 2018; Adi et al., 2018; Merrer et al., 2017; Dong et al., 2018; Szyller et al., 2019). We compare our DNN fingerprinting to another proposed DNN fingerprint called IPGuard (Cao et al., 2019). We refer to Appendix A.1 for more details on the CIFAR-10 experiments and to Appendix A.2 for results on ImageNet32.

**Evaluation Metrics.** We measure the fingerprint retention as the success rate of conferrable examples in a target model $\hat{M}$. An example is successful if the target model predicts the target label

| $\epsilon$ | 0.01 | 0.025 | 0.05 | 0.075 | 0.1 | 0.125 | 0.15 |
|---|---|---|---|---|---|---|---|
| $\theta_\epsilon$ | 63.00% | 75.00% | 84.00% | 86.00% | 87.00% | 87.00% | 87.00% |

Table 1: Experimentally derived values for the decision threshold $\theta_\epsilon$ for CIFAR-10.

given in $\mathcal{F}_y$. We refer to this metric as the *Conferrable Adversarial Example Accuracy* (CAEAcc).

$$\text{CAEAcc}(\hat{M}(\mathcal{F}), \mathcal{F}_y) = \Pr_{(y,y^*)\in\hat{M}(\mathcal{F}),\mathcal{F}_y}\left[\mathbf{1}_{\text{y}=\text{y}^*}\right] \tag{7}$$

**CIFAR-10 models.** We train popular CNN architectures on CIFAR-10 without any modification to the standard training process[2]. The defender's source model has a ResNet20 (He et al., 2016) architecture. All surrogate and reference models required by the defender for CEM are ResNet20 models trained on CIFAR-10 using retraining. The attacker has access to various model architectures, such as DenseNet (Iandola et al., 2014), VGG-16/VGG-19 (Simonyan & Zisserman, 2014) and ResNet20 (He et al., 2016), to show that conferrability is maintained across model architectures.

**Removal Attacks.** Removal attacks are successful if (i) the surrogate model has a high test accuracy (at least $85.55\%$ for CIFAR-10, see Fig. 4e) and (ii) the stolen surrogate's CAEAcc is lower than the verification threshold $\rho$. Stolen models are derived with a wide range of fingerprint removal attacks, which we categorize into (i) model modification, (ii) model extraction and (iii) adapted model extraction attacks. Model modification attacks modify the source model directly, such as fine-tuning or parameter pruning. Model extraction distills knowledge from a source model into a fresh surrogate model, such as retraining or model extraction attacks from related work (Papernot et al., 2017; Jagielski et al., 2019; Orekondy et al., 2019). A defense against model extraction attacks is to restrict the source model's output to the top-1 predicted label. We refer to an extraction attack that retrains on the top-1 predicted label by the source model as *Black-Box* attack. We also evaluate transfer learning as a model extraction attack, where a pre-trained model from a different domain (ImageNet32) is transfer learned onto the source model's domain using the source model's labels.

Adapted model extraction attacks limit the transferability of adversarial examples in the surrogate model, such as adversarial training (Madry et al., 2017). In our experiment, we use multi-step adversarial training with PGD. We design another adapted attacks against our fingerprint called the *Ground-Truth* attack, where the attacker has a fraction of $p \in [0.6, 0.7, 0.8]$ ground-truth labels. We trained surrogate models with Differential Privacy (DP), using DP-SGD (Abadi et al., 2016), but even for large $\epsilon$ the surrogate models had unacceptably low CIFAR-10 test accuracy of about $76\%$. The test accuracies of all surrogate models can be seen in Table 2. We repeat all removal attacks three times and report the mean values and the standard deviation as error bars. Our empirically chosen CIFAR-10 decision thresholds $\theta_\epsilon$ (see Table 1) are chosen relative to the perturbation threshold $\epsilon$ with which our adversarial attacks are instantiated.

**Evasiveness Attack.** Hitaj et al. (2019) present an evasion attack against black-box watermark verification. Their evasion attack trains a binary DNN classifier to distinguish between benign and out-of-distribution images and reject queries by the defender. Our attacker trains a binary classifier to classify benign images and adversarial examples generated by FGM, CW-$L_\infty$, and PGD adversarial attacks. The evasion attack is evaluated by the area under the curve (AUC) of the receiver operating characteristic (ROC) curve for varying perturbation thresholds.

**Attacker Datasets.** We distinguish between attackers with access to different datasets, which are CIFAR-10 (Krizhevsky et al.), CINIC (Darlow et al., 2018) and ImageNet32 (Chrabaszcz et al., 2017). CINIC is an extension of CIFAR-10 with downsampled images from ImageNet (Deng et al., 2009), for classes that are also defined in CIFAR-10. ImageNet32 is a downsampled version of images from all classes defined in ImageNet, i.e., ImageNet32 is most dissimilar to CIFAR-10. We observe that the similarity of the attacker's to the defender's dataset positively correlates with the surrogate model's test accuracy and thus boosts the effectiveness of the model stealing attack.

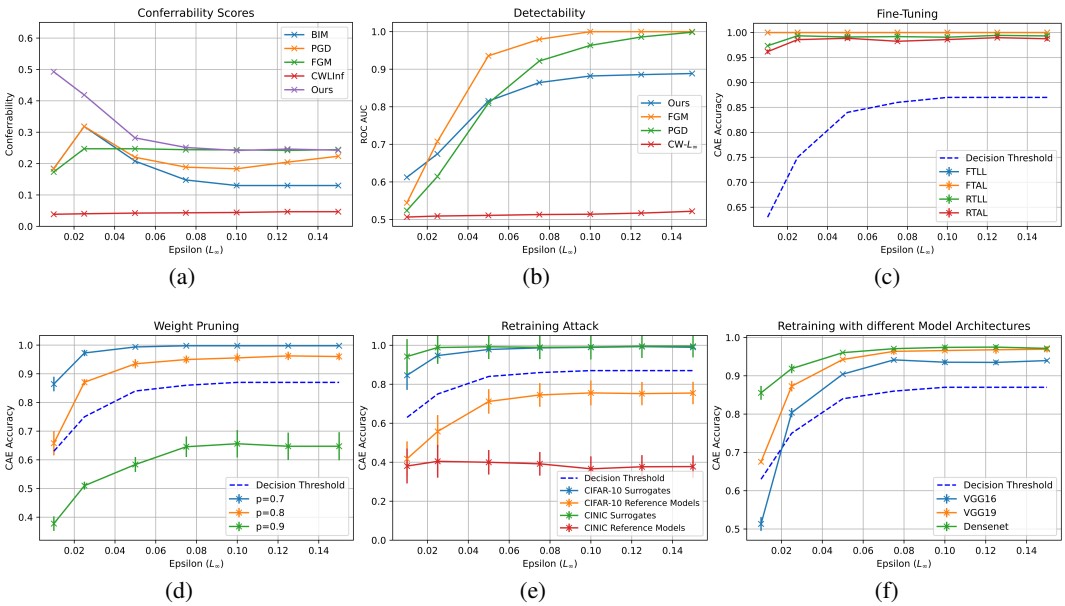

Figure 3: Results for the (a) conferrability scores, (b) non-evasiveness and (c-d) irremovability to model modification attacks. Figures (e,f) show results against model extraction attacks.

## 6 EMPIRICAL RESULTS

**Conferrability Scores.** Our empirical results show support in favor of the existence of conferrable adversarial examples. CEM produces adversarial example with the highest conferrability scores compared to classical adversarial attacks such as FGM, PGD and CW-$L_\infty$. We generate $n = 100$ adversarial examples from the same set of CIFAR-10 test samples, filter those that are non-successful to the source model and evaluate conferrability scores as specified by Equation 2 across five unseen CIFAR-10 surrogate and reference models trained from scratch. The results in Fig. 3a show that CEM outputs adversarial examples with significantly higher conferrability scores than other attacks for small perturbation thresholds $\epsilon$, which worsen as $\epsilon$ increases. For $\epsilon = 0.01$ we measure a mean conferrability score of $0.49$, which amounts to a mean CAEAcc of $0.42$ in reference models and $0.85$ in surrogate models (see Fig. 3d). For $\epsilon = 0.15$ we measure a conferrability score of only $0.24$, which translates to a mean CAEAcc of $0.98$ in surrogate models and $0.75$ in reference models. These results show that with increasing $\epsilon$, the transferability of generated adversarial examples improves at the expense of lower conferrability. We observed that the conferrability scores measured on surrogate and reference models used in CEM are nearly perfect. This indicates that CEM could benefit from access to an even larger ensemble of models.

**Non-Evasiveness.** We evaluate non-evasiveness against the attack proposed by Hitaj et al. (2019). The results in Fig. 3b show that (i) detectability increases with larger perturbation thresholds $\epsilon$ and (ii) for small perturbation thresholds $\epsilon \leq 0.025$ we measure a ROC AUC of only $0.67$. This method of detection is too unreliable for the attacker to deploy in practice considering that only a fraction of requests would contain the fingerprint. Sufficiently high recall against our fingerprint comes at the cost of more false positives, which diminishes the utility of the attacker's model to other users.

**Model Modification Attacks.** Fig. 3c and 3d show that our fingerprint is robust against model modification attacks. We show irremovability against four different types of fine-tuning attacks tried by Uchida et al. (2017), which are implemented as follows.

1. Fine-Tune Last Layer (**FTLL**): Freezes all layers except for the penultimate layer.
2. Fine-Tune All Layers (**FTAL**): Fine-tuning of all layers.

---

[2]https://keras.io/examples/cifar10_resnet

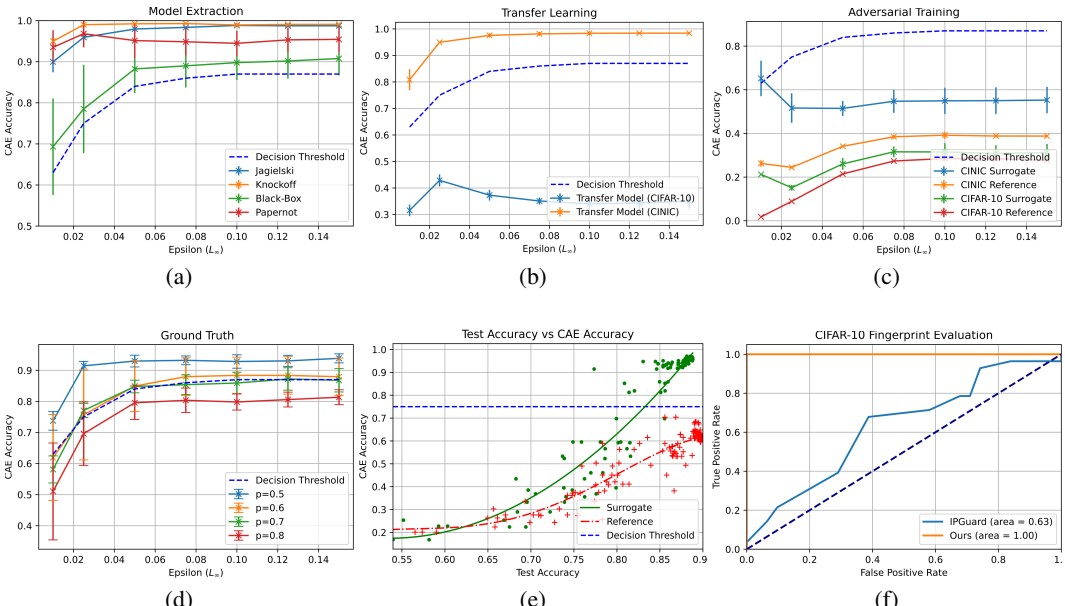

Figure 4: Fig. (a-d) show the robustness of our fingerprint against (adapted) model extraction attacks. Fig. (e) shows the test accuracy in relation to the CAEAcc for surrogate and reference models during training. Fig. (f) shows the ROC curve of our fingerprint compared to IPGuard (Cao et al., 2019).

3. Retrain Last Layer (**RTLL**): Re-initializes the penultimate layer's weights and updates only the penultimate layer, while all other layers are frozen.

4. Retrain All Layers (**RTAL**): Re-initializes the penultimate layer's weights, but all layers are updated during fine-tuning.

Our results for iterative weight pruning (Zhu & Gupta, 2017) show that larger pruning rates $p \in [0.7, 0.8, 0.9]$ have greater impact on the CAEAcc, but the surrogate test accuracy also deteriorates significantly (see Table 2). Note that we show robustness to much higher pruning rates than related work (Cao et al., 2019). The attack for $p = 0.9$ is unsuccessful because the surrogate test accuracy is lower than $85.55\%$, as described in Section 5. For all remaining pruning configurations the fingerprint is not removable.

**Model Extraction Attacks.** Fig. 3 and 4 show that our fingerprint is robust to model extraction attacks, except for transfer learning when the attacker has CIFAR-10 data. Surprisingly, we find that surrogates extracted using CINIC have higher mean CAEAcc values than CIFAR-10 surrogates at a lower test accuracy ($-2.16\%$). Similarly, we measure significantly lower CAEAcc values for reference models trained on CINIC compared to reference models trained on CIFAR-10. This means that it is increasingly difficult for the attacker to remove our fingerprint the more dissimilar the attacker's and defender's datasets become. In Fig. 3f, we show that our fingerprint is not removable even across different surrogate model architectures. Fig. 4b shows that transfer learning removes our fingerprint when the attacker has access to the defender's dataset, but our fingerprint is not removable when the attacker only has access to CINIC data. We observed that the pre-trained ImageNet32 model exceeds 80% test accuracy after only a single epoch with CIFAR-10, which may be too few updates for our fingerprint to transfer to the surrogate. The irremovability of our fingerprint to a wide range of model extraction attacks shows that certain adversarial vulnerabilities (that enable conferrable examples) are consistently carried over from the source model to its surrogates. Our data supports that the class of transferable examples can be broken down further into conferrable examples and that known findings for transferable examples extend to conferrable examples.

**Adapted Model Extraction Attacks.** We now show limitations of our fingerprint against defenses that specifically limit the transferability of adversarial examples. Fig. 4c shows that adversarial

| | Fine-Tuning | | | | Pruning | | |
|---|---|---|---|---|---|---|---|
| Source | FTLL | FTAL | RTLL | RTAL | p=0.7 | p=0.8 | p=0.9 |
| 89.30 | 89.59 | 89.57 | 87.90 | 87.25 | 88.00 | 86.97 | 83.36 |
| Retrain | | | | | Extraction | | |
| ResNet20 | Densenet | VGG16 | VGG19 | CINIC | Jagielski | Papernot | Knockoff |
| 89.22 | 90.88 | 90.91 | 90.05 | 87.06 | 88.74 | 87.34 | 84.55 |
| AdvTrain | | Transfer Learning | | | Ground-Truth | | |
| CIFAR-10 | CINIC | CIFAR-10 | CINIC | p=0.5 | p=0.6 | p=0.7 | p=0.8 |
| 89.75 | 88.15 | 89.59 | 88.15 | 89.54 | 89.13 | 89.43 | 89.33 |

Table 2: Mean CIFAR-10 test accuracies for surrogate models obtained by running the attacks with threefold repetition. Unless stated otherwise, all surrogates are trained on CIFAR-10 images.

training from scratch removes our fingerprint. In CIFAR-10 surrogates, we measure a mean CAEAcc of only $15\%$ for $\epsilon = 0.025$, which supports the claims by Madry et al. (2017) that adversarial training increases robustness to transfer attacks. We hypothesize that incorporating adversarial training into the generation process of conferrable adversarial examples leads to higher robustness against this removal attack. The results from our Ground-Truth attack shown in Fig. 4d show that access to more ground-truth labels decreases the CAEAcc in surrogate models. The results show that our fingerprint is not removable for attackers with up to $50\%$ ground-truth labels for CIFAR-10.

**Confidence Analysis.** In Fig. 4f we show the ROC curve (false positive vs true positive rate) of our fingerprint verification on ten unseen, retrained CIFAR-10 surrogate and reference models. We compare our work with IPGuard (Cao et al., 2019) and generate $n = 100$ fingerprints in both cases. Our ROC AUC is $1.0$, whereas IPGuard has a ROC AUC of only $0.63$. These results show that IPGuard is not robust to retraining as a model extraction attack. Inspecting adversarial examples generated by IPGuard further, we measure a mean adversarial success rate of $17.61\%$ for surrogate and $16.29\%$ for reference models. For our fingerprint, Fig. 4e visualizes the difference in CAEAcc for well-trained surrogate and reference models at $\epsilon = 0.025$. The plot shows that CAEAcc positively correlates with the surrogate's test accuracy. We measure a mean difference in CAEAcc of about $30\%$ between well-trained surrogate and reference models with our fingerprint.

## 7 CONCLUSION

We empirically show the existence of conferrable adversarial examples. Our ensemble adversarial attack CEM outperforms existing adversarial attacks such as FGM (Goodfellow et al., 2014), PGD (Madry et al., 2017) and CW-$L_\infty$ (Carlini & Wagner, 2017b) in producing highly conferrable adversarial examples. We formally define fingerprinting for DNN classifiers and use the generated conferrable adversarial examples as our fingerprint. Our experiments on the robustness of our fingerprint show increased robustness to model modification attacks and model extraction extraction attacks. Transfer learning is a successful removal attack when the attacker has access to CIFAR-10 data, but not when the attacker only has access to CINIC. Adversarial training from scratch is the most effective removal attack and successfully removes our fingerprint. We hypothesize that adding adversarial training into the generation process of conferrable adversarial examples may increase robustness against adversarial training. Our experiments confirm the non-evasiveness of our fingerprint against a detection method proposed by Hitaj et al. (2019). We empirically find that our fingerprint is the first to perfectly verify retrained CIFAR-10 surrogates with a ROC AUC of 1.0.

## 8 ACKNOWLEDGEMENTS

We gratefully acknowledge the support of NSERC for grants RGPIN-05849, CRDPJ-531191, IRC537591, and the Royal Bank of Canada for funding this research.

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

# A APPENDIX

## A.1 SUPPLEMENTARY MATERIAL FOR CIFAR-10 EXPERIMENTS

**Setup.** We train all models on a server running Ubuntu 18.04 in 64-bit mode using four Tesla P100 GPUs and 128 cores of an IBM POWER8 CPU (2.4GHz) with up to 1TB of accessible RAM. The machine learning is implemented in Keras using the Tensorflow v2.2 backend, and datasets are loaded using Tensorflow Dataset[3]. We re-use implementations of existing adversarial attacks from the Adversarial Robustness Toolbox v1.3.1 (Nicolae et al., 2018). DP-SGD (Abadi et al., 2016) is implemented through Tensorflow Privacy (Galen Andrew, Steve Chien, and Nicolas Papernot, 2019). All remaining attacks and IPGuard (Cao et al., 2019) are re-implemented from scratch.

For the generated adversarial examples using attacks like FGM, PGD and CW-$L_\infty$, we use the following parametrization. We limit the maximum number of iterations in PGD to $10$ and use a step-size of $0.01$. For FGM, we use a step-size of $\epsilon$ and for CW-$L_\infty$ we limit the number of iterations to $50$, use a confidence parameter $k = 0.5$, a learning rate of $0.01$ and a step-size of $0.01$. Fingerprints generated by CEM for CIFAR-10 can be seen in Fig. 5a.

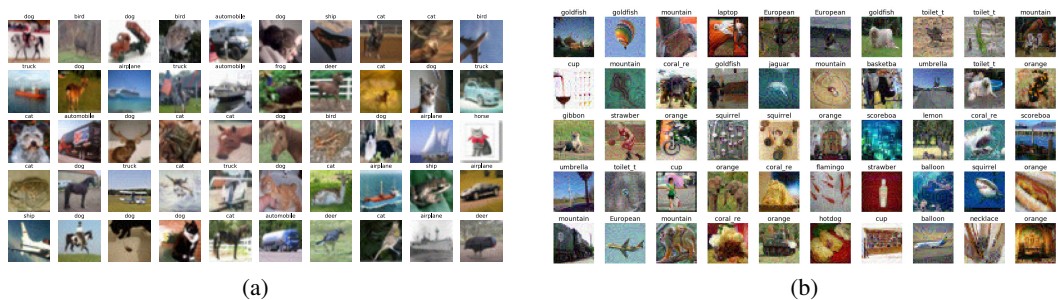

(a)                                          (b)

Figure 5: Fig. (a) shows conferrable examples generated with CEM on CIFAR-10 with $\epsilon = 0.025$. Fig. (b) shows conferrable examples generated with CEM on ImageNet32 with $\epsilon = 0.1$.

**Analyzing Conferrable Examples.** We further examine our fingerprint to find an explanation for the large mean CAE accuracy in all reference models of more than $50\%$, even though the baseline for random guessing is just $10\%$. For this, we plot the confusion matrix for our fingerprint at $\epsilon = 0.025$ in Fig. 6.

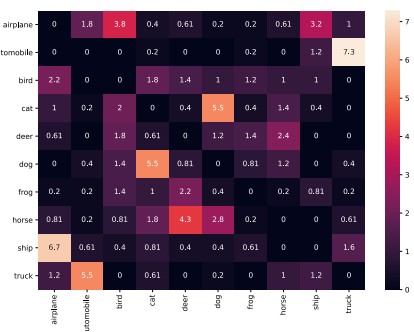

Figure 6: Confusion matrix for our CIFAR-10 fingerprint at $\epsilon = 0.025$ with the initial label on the vertical axis and the target label on the horizontal axis. Depicted values are normalized and represent percentages.

The confusion matrix shows that all classes are represented in the fingerprint and the target class distribution is balanced. We notice a symmetry in the confusion matrix along the diagonal line. A

---

[3]https://www.tensorflow.org/datasets

source-target class pair appears more often in our fingerprint if the classes are more similar to each other, e.g. classes 'dog' and 'cat' or 'automobile' and 'truck'. We hypothesize that this partially explains why reference models have large CAE accuracies of over $50\%$, even though the baseline for random guessing would be at $10\%$. A reference model is more likely to misclassify the class 'cat' as the class 'dog' than any other class, which moves the baseline closer to $50\%$ for many cases. The similarity of the reference model to the source model is another important aspect to explaining the high CAE accuracy in the reference model. Our experiments show that reference models trained on the same dataset are also more likely to share the same adversarial vulnerabilities.

## A.2 EXPERIMENTS ON IMAGENET32

We select images from the following 100 classes of ImageNet32:

'kit fox', 'Persian cat', 'gazelle', 'porcupine', 'sea lion', 'killer whale', 'African elephant', 'jaguar', 'otterhound', 'hyena', 'sorrel', 'dalmatian', 'fox squirrel', 'tiger', 'zebra', 'ram', 'orangutan', 'squirrel monkey', 'komondor', 'guinea pig', 'golden retriever', 'macaque', 'pug', 'water buffalo', 'American black bear', 'giant panda', 'armadillo', 'gibbon', 'German shepherd', 'koala', 'umbrella', 'soccer ball', 'starfish', 'grand piano', 'laptop', 'strawberry', 'airliner', 'balloon', 'space shuttle', 'aircraft carrier', 'tank', 'missile', 'mountain bike', 'steam locomotive', 'cab', 'snowplow', 'bookcase', 'toilet seat', 'pool table', 'orange', 'lemon', 'violin', 'sax', 'volcano', 'coral reef', 'lakeside', 'hammer', 'vulture', 'hummingbird', 'flamingo', 'great white shark', 'hammerhead', 'stingray', 'barracouta', 'goldfish', 'American chameleon', 'green snake', 'European fire salamander', 'loudspeaker', 'microphone', 'digital clock', 'sunglass', 'combination lock', 'nail', 'altar', 'mountain tent', 'scoreboard', 'mashed potato', 'head cabbage', 'cucumber', 'plate', 'necklace', 'sandal', 'ski mask', 'teddy', 'golf ball', 'red wine', 'sunscreen', 'beer glass', 'cup', 'traffic light', 'lipstick', 'hotdog', 'toilet tissue', 'cassette', 'lotion', 'barrel', 'basketball', 'barbell', 'pole'

We train a ResNet20 source model and locally retrain $c_1 = 14$ surrogate and $c_2 = 15$ reference models. In contrast to the experiments on CIFAR-10, we allow the defender access to various model architectures, such as ResNet56, Densenet, VGG19 and MobilenetV2. We want to verify if the generation of conferrable adversarial examples is feasible when the defender trains a set of models with larger variety in model architecture. In practice, it may be the case that the defender locally searches for the best model architecture and in the process obtains multiple surrogate and reference models with various model architectures.

Table 3 and 4 show the test accuracies and CAEAcc values of ImageNet32 surrogate and reference models. Values in the brackets denote the lowest and highest value measured. Our results for ImageNet32 are comparable to the results obtained with models trained on CIFAR-10. We even measure a lower CAE accuracy in reference models than we did in reference models trained on CIFAR-10. Note that the experiments conducted on ImageNet32 are results reported for $\epsilon = 0.15$ and the results for CIFAR-10 were generated with a perturbation threshold of $\epsilon = 0.025$.

## A.3 FINGERPRINTING DEFINITIONS

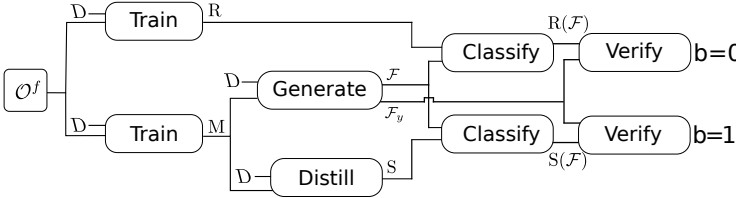

Figure 7: A schematic illustration of the source model and the two types of models, the surrogate $S$ and the reference model $R$, that a fingerprint verification should distinguish. 'Distill' is any distillation attack that results in a surrogate model with similar performance as the source model and 'Classify' returns the output of a model on a set of inputs.

In this section, we formally define the term 'surrogate' and provide security games for *irremovability* and *non-evasiveness* of DNN fingerprinting. For ease of notation, we define an auxiliary function to

| | | Fine-Tuning | | | |
|---|---|---|---|---|---|
| Attack | Source | FTLL | FTAL | RTLL | RTAL |
| Test Acc | 55.70 | 57.30 | 61.30 | 59.00 | 45.00 |
| CAEAcc | 100.00 | 100.00 | 100.00 | 100.00 | 82.00 |
| | | Retrain | | | |
| Attack | ResNet20 | ResNet56 | Densenet | VGG19 | MobileNetV2 |
| Test Acc | 53.10 | 54.50 | 50.95 | 52.50 | 52.87 |
| CAEAcc | 97.00 | 99.00 | 90.00 | 76.00 | 97.00 |
| | | Extraction | | | |
| Attack | Jagielski | Papernot | Knockoff | | |
| Test Acc | 53.20 | 50.90 | 47.40 | | |
| CAEAcc | 98.00 | 90.00 | 98.00 | | |

Table 3: Surrogate model accuracies for ImageNet32.

| | ResNet20 | ResNet56 | Densenet | VGG19 | MobileNetV2 |
|---|---|---|---|---|---|
| Test Acc | 55.00 | 59.10 | 55.95 | 54.10 | 62.90 |
| CAEAcc | 47.00 | 50.00 | 47.00 | 34.00 | 61.00 |

Table 4: Reference model accuracies for ImageNet32.

generate a fingerprint for a source model.

FModel():

1. Compute $M \leftarrow \text{Train}(\mathscr{O}, D)$
2. Sample $(\mathcal{F}, \mathcal{F}_y) \leftarrow \text{Generate}(M, D)$
3. Output $(M, \mathcal{F}, \mathcal{F}_y)$

**Definition of Surrogates.** We define the randomized process 'distill' that on the input of a source model $M$ and a dataset $D \in \mathcal{D}$, returns a distilled surrogate $S$ of the source model. Note that other common methods of derivation, such as retraining, fine-tuning, and pruning on labels provided by the source model, can be expressed in terms of knowledge distillation (Hinton et al., 2015).

For a given source model $M$ we recursively define the set $\mathcal{S}_M$ of its *surrogate* models as follows.

$$S_M \leftarrow \{\text{distill}(S, D) | S \in \mathcal{S}_M \cup \{M\}\} \tag{8}$$

If a model $R \in \mathcal{M}$ is trained from a labeled dataset without access to $\mathcal{S}$ and hence is not contained in the set of surrogate models for source model $M$, we refer to that model as *reference model* relative to $M$. The goal of this paper is to find a method that verifies whether a surrogate model is in $\mathcal{S}_M$.

**Irremovability.** The attacker accesses the source model $M$ to derive a surrogate $\hat{M} \leftarrow \mathcal{A}(M)$. We say fingerprinting is not removable if the adversary has a low probability of winning the following security game.

1. Defender computes $(M, \mathcal{F}, \mathcal{F}_y) \leftarrow \text{FModel}()$
2. Obtain $\hat{M}_0 \leftarrow \text{Train}(\mathscr{O}, \mathcal{D})$ and $\hat{M}_1 \leftarrow \mathcal{A}(M)$
3. Sample $b \xleftarrow{\$} \{0, 1\}$ and send $\hat{M}_b$ to the Defender
4. Adversary wins if:

$$Pr[\text{Verify}(\hat{M}_b(\mathcal{F}), \mathcal{F}_y) = b] \approx 0.5$$

Note that our definition of a robust fingerprint differs from related work (Cao et al., 2019) because we include in the set of removal attacks $\mathcal{A}$ also model extraction attacks. Fig. 7 schematically shows the types of models that a fingerprint verification should distinguish.

**Non-Evasiveness.** Hitaj & Mancini (2018) show that an attacker can evade the black-box verification by rejecting queries or returning random labels when the verification process is detectable. We

specify a *non-evasiveness* property, which makes it hard to separate members of the fingerprint to benign data samples. A fingerprinting method is non-evasive, if for some $\varepsilon \in \mathbb{R}^+$, the adversary has a low probability of winning the following security game. Let $\text{Adv}(\mathcal{D}, \epsilon)$ be an adversarial attack with maximum perturbation $\epsilon \in \mathbb{R}^+$ performed on the dataset $D$. An adversarial example with a perturbation smaller than $\epsilon$ can be hard to detect (Carlini & Wagner, 2017a). The security game is defined as follows.

1. Compute $(M, \mathcal{F}, \mathcal{F}_y) \leftarrow \text{FModel}()$

2. Adversary trains detector $\hat{M} \leftarrow \mathcal{A}(M)$

3. Sample $b \xleftarrow{\$} \{0, 1\}$

4. Set $X_0 \xleftarrow{\$} \text{Adv}(D, \epsilon)$ and $X_1 \xleftarrow{\$} \mathcal{F}$ and send $X_b$ to the adversary

5. Adversary wins if

$$Pr[\hat{M}(X_b) = b] > 0.5 + \varepsilon$$

Note that $\varepsilon$ bounds the trade-off between evasiveness and the suspect model's performance. When $\varepsilon$ becomes smaller, more queries from benign users are falsely rejected, or a random label is returned at the same level of evasiveness. The goal of the attacker is to find a model that maximizes evasiveness while also maximizing model performance.

## A.4 OPTIMIZING CONFERRABILITY

In this section, we formally present the optimization problem that has to be solved to find highly conferrable adversarial examples. We denote the set of surrogate models for a source model $M$ by $\mathcal{S}_M$ and the set of reference models as $\mathcal{R}$.

Conferrability scores should be minimal when all reference and surrogate models assign the same target label (perfect transferability), or none of them assign the target label (non-transferability). An adversarial example that is perfectly transferable to surrogate models and non-transferable to reference models should have the maximum conferrability score. The following formula satisfies these constraints for a target class $t \in \mathcal{Y}$.

$$\text{Confer}(\mathcal{S}, \mathcal{R}, x; t) = \text{Transfer}(\mathcal{S}, x; t)(1 - \text{Transfer}(\mathcal{R}, x; t)) \tag{9}$$

Note that Equation 9 does not have weight factors, as it is equally important to transfer to surrogate models as it is to not transfer to reference models.

The optimization constraints to find adversarial examples with high conferrability scores can be formalized as follows for a benign input $x \in \mathcal{X}$ and a perturbation $\delta$ in the infinity norm.

Minimize $\delta$ s.t.
1. $M(x + \delta) = t$
2. $\Pr_{S \in \mathcal{S}}[S(x + \delta) = t] \approx 1$
3. $\Pr_{R \in \mathcal{R}}[R(x + \delta) \neq t] \approx 1$

subject to $||\delta||_\infty \leq \epsilon$

The challenge of generating conferrable examples is to find a good optimization strategy that (i) finds local minima close to the global minimum and (ii) uses the least amount of surrogate and reference models to find those conferrable examples. Obtaining many surrogate and reference models that allow optimizing for conferrability is a one-time effort, but may be a practical limitation for datasets where training even a single model is prohibitively expensive.

## A.5 REMOVAL ATTACKS

In this section, we provide more details on the removal attacks and their parametrization.

| $\alpha$ | $\beta$ | $\gamma$ | **Confer** | $\alpha$ | $\beta$ | $\gamma$ | **Confer** |
|---|---|---|---|---|---|---|---|
| 0.1 | 0.1 | 0.5 | 0.20 | 0.5 | 1.0 | 0.1 | 0.44 |
| 0.1 | 0.1 | 1.0 | 0.20 | 0.5 | 1.0 | 0.5 | 0.37 |
| 0.1 | 0.5 | 0.1 | 0.20 | 0.5 | 1.0 | 1.0 | 0.29 |
| 0.1 | 0.5 | 0.5 | 0.20 | 1.0 | 0.1 | 0.1 | **0.49** |
| 0.1 | 0.5 | 1.0 | 0.20 | 1.0 | 0.1 | 0.5 | 0.42 |
| 0.1 | 1.0 | 0.1 | 0.20 | 1.0 | 0.1 | 1.0 | 0.40 |
| 0.1 | 1.0 | 0.5 | 0.20 | 1.0 | 0.5 | 0.1 | 0.45 |
| 0.1 | 1.0 | 1.0 | 0.20 | 1.0 | 0.5 | 0.5 | 0.37 |
| 0.5 | 0.1 | 0.1 | 0.20 | 1.0 | 0.5 | 1.0 | 0.37 |
| 0.5 | 0.1 | 0.5 | 0.20 | 1.0 | 1.0 | 0.1 | 0.41 |
| 0.5 | 0.1 | 1.0 | 0.20 | 1.0 | 1.0 | 0.5 | 0.42 |
| 0.5 | 0.5 | 0.1 | 0.20 | 1.0 | 1.0 | 1.0 | 0.40 |
| 0.5 | 0.5 | 1.0 | 0.20 | | | | |

Table 5: An empirical sensitivity analysis of the hyperparameters in Eq. 6. Confer measures the mean conferrability score over five surrogate and five reference models trained on CIFAR-10.

### A.5.1 MODEL EXTRACTION

Retraining uses the source model $M$ to provide labels to the training data $D$ and then uses these labels to train the surrogate model $S$.

$$S \leftarrow \text{Train}(M, D) \tag{10}$$

Jagielski et al. (2019) post-process the labels received from the source model by a distillation parameter $T'$ to obtain soft labels. For an input $x \in \mathcal{D}$ and a source model $M$, the soft labels $M'(x)$ can be computed as follows.

$$M'(x)_i = \frac{\exp(M(x)_i^{1/T'})}{\sum_j \exp(M(x)_j^{1/T'})} \tag{11}$$

The surrogate model is trained on the soft labels.

$$S \leftarrow \text{Train}(M', D) \tag{12}$$

Papernot et al. (2017) propose training the surrogate model iteratively starting with an initial dataset $D_0$ that is concatenated after each round with a set of the surrogate model's adversarial examples.

$$S_0 \leftarrow \text{Train}(M, D_0) \tag{13}$$
$$D_{i+1} \leftarrow \{x + \lambda \cdot \text{sign}(J_S[M(x)]) | x \in D_i\} \cup D_i \tag{14}$$
$$S_i \leftarrow \text{Train}(M, D_{i+1}) \tag{15}$$

$J_S$ denotes the Jacobian matrix computed on the surrogate model $S$ for the labels assigned by the source model $M$. We use the FGM adversarial attack to generate adversarial examples, as described by the authors.

Knockoff (Orekondy et al., 2019) uses cross-domain transferability to derive a surrogate model. They construct a *transfer set*, which is cross-domain data, used to derive the surrogate model. We implement the random selection approach presented by the authors

### A.6 EMPIRICAL SENSITIVITY ANALYSIS

We perform a grid-search to understand the sensitivity of the choice of hyperparameters in Eq. 6 on the mean conferrability score over $n = 50$ inputs. Mean conferrability is evaluated over five surrogate and five reference models trained on CIFAR-10, which have not been used in the generation process of the conferrable examples. We evaluate all parameters in the range $[0.1, 0.5, 1.0]$ and leave out all combinations where $\alpha = \beta = \gamma$, except for the baseline used in this paper ($\alpha = \beta = \gamma = 1$). The examples are generated for $\epsilon = 0.025$ with an Adam optimizer (Kingma & Ba, 2014) with a learning rate $5e-4$ and we optimize for 300 iterations.

The results are illustrated in Table 5. We find that when $\alpha$ is small, the optimization does not converge and the source model does not predict the target label. When $\alpha$ is large relative to the other parameters, we measure the highest mean conferrability. For $\alpha = 1.0$ and $\beta = \gamma = 0.1$ we measure a mean conferrability of 0.49, which improves over the paper's baseline.

