# OpenReview forum: "Deep Neural Network Fingerprinting by Conferrable Adversarial Examples"
_ICLR.cc/2021/Conference — ICLR 2021 Spotlight_

### Official Review · AnonReviewer4 · 2020-10-27
**An interesting paper.**

**Rating:** 6
**Confidence:** 2

**Review:**

Summary:
This paper introduces an interesting property of adversarial examples, which is called conferrability and can reflect the abilities whether an instance can exclusively transfer with a target label from a source model to its surrogates. A new method is proposed to generate conferrable adversarial examples. Experimental results show the effectiveness of the proposed method. The most impressive thing is the AUC of this method in verifying surrogates.

Comments:

I'm not familiar enough with the field to judge the novelty and soundness of the proposed method. But I'm willing to roll with the main idea of the paper. Conferrability is the core contribution of the paper. The existence of the conferrable adversarial examples is shown empirically in the paper. Ensemble adversarial attack CEM is shown effective in the same time.

Strengths:

1.The research on this issue has extremely high application value and practical significance. The problem is very important for providers, especially when they want to protect their intellectual property.

2.This paper is well motivated, and thus it is enjoyable to read.

3.The idea of conferrability is very interesting. The existence of the conferrable adversarial examples is shown empirically to prove the value of this definition.

4.Impressive empirical results. Fingerprint in this paper is the first method that reaches an AUC of 1.0 in verifying surrogates, which is impressive for me. Because I'm not familiar enough with the field, I'm not sure whether the method is SOTA in more benchmarks.

5.Interesting figures make the paper easy to follow. I suggest the authors replace Figure 2(b) with a vetorgram to improve the quality.

Weaknesses:

1.There are 3 hyperparameters in equation 5, sensitivity analysis may be needed.

2.Is decision threshold sensitive in different datasets? It may affect the practicality of the method.

3.Clarity: What is the meaning of $\sigma$ in equation (4)? Is it an activation function? More clarification may be needed to make the paper easier to read.

4.What is the meaning of 'unremovable'? Is this a formal term in English (irremovable)?

---

> ### Author Response · Authors · 2020-11-20
> **All changes have been incorporated**
>
> We thank the reviewer for their constructive feedback and their interest in our paper.
> Our updated revision addresses the reviewer's comments as follows.
>
> We have added a preliminary empirical local sensitivity analysis for the three hyperparameters from Formula 6 in the Appendix (A.6).
> However, we believe a more extensive analysis is part of future work.
> We would be happy to receive comments from the reviewer on the analysis.
>
> In our experiments, the decision thresholds between all evaluated datasets (CIFAR-10 and ImageNet32) do not differ (see Appendix 3).
> In any case, the defender has the ability to derive the decision threshold from the fingerprint generation process.
>
>
> We have used the additional space to improve clarity and made the following updates to the paper.
>
> - We clarified that $\sigma$ refers to the Softmax activation function.
>
> - Figure 2 (b) is now vectorized.
>
> - We are not native speakers of the English language and re-used the word "unremovability" defined by Adi et. al [1].
> The reviewer's suggestion is correct and we refer to this property as "irremovability" in the updated submission, which is a defined noun in the dictionary [2].
>
> -----------------
>
> [1] Adi, Yossi, et al. "Turning your weakness into a strength: Watermarking deep neural networks by backdooring." 27th {USENIX} Security Symposium ({USENIX} Security 18). 2018.
>
> [2] https://www.merriam-webster.com/dictionary/irremovability

---

### Official Review · AnonReviewer2 · 2020-10-28
**the threat model is unclear**

**Rating:** 6
**Confidence:** 3

**Review:**

This paper proposed a fingerprinting approach for identifying stolen models. To distinguish stolen models from reference models, the proposed approach generated conferrable adversarial examples, which can only be transferred to the stolen models, but not to the reference models.

Pros:
1. The idea of using adversarial examples to identify stolen models is interesting.
2. The paper provides comprehensive experiments, including model extraction attacks, adaptive model extraction attacks, etc.  The results outperform popular adversarial attacks, FGM, PGD, and CW attacks.

Cons:
1. The key concern about the paper is the unclear threat model. Why does the model stealing attacker have white-box access to the source model? In general, the white-box access means the attacker has all the information about the source model. The definition of “strong attacker” is confusing: If the attacker requires access to domain data, then why the attacker is strong? Do attackers have access to the data? It seems the attackers have only the input data but partial label data, which is a strong assumption. Many recent works show that model stealing attacks can surrogate datasets to extract the victim models. Can these attacks be detected by the proposed approach?
2. The paper is very hard to follow. Many definitions are missing in the paper.
3. What is Transfer(S, x; t) in Eq (1)? What is Classify(S, x) in Eq (2) and (3)? What is H in Eq (5)?
4. In the conclusion section, CW-L2 should be L-infinity?

---

> ### Author Response · Authors · 2020-11-20
> **Updated threat model**
>
> We sincerely thank the reviewer for their valuable feedback and apologize for the confusion the term 'strong attacker' has caused.
> Our updated threat model now refers to an 'informed attacker' in favor of a 'strong attacker'.
> A more informed attacker can perform at least the same (or more) attacks than a less informed attacker.
> The more informed attacker can drop information and invoke all attacks of a less informed attacker.
> For example, an attacker with white-box access is more informed than an attacker with black-box access and can invoke their attacks.
> We evaluate our fingerprint against differently informed attackers and show robustness even against highly informed attackers (e.g., white-box, partially labeled data), which implies robustness against less informed attackers (e.g., black-box, no data labels).
>
> In our evaluation, we experiment with three different types of datasets that an attacker could use to remove the fingerprint.
> These are (i) unlabeled data from a different distribution, (ii) unlabeled data from the same distribution (i.e., domain data), and (iii) partially labeled data from the same distribution.
> Our evaluation shows that all of these attacks are detectable.
> Access to (labeled) data from the same distribution improves the attack outcomes (i.e., higher test accuracy, lower mean CAEAcc).
> This is in line with our attacker model, where a more informed attacker can perform more attacks than a less informed attacker.
> This phenomenon is described in more detail on page 7, "Model Extraction Attacks".
>
> We use the extra page to improve the clarity of the paper's presentation and add the requested definitions.
> Here is a brief answer to the raised questions.
>
> - Transfer(M, x; t) is the probability that input x transfers with a target class t to models from the set M. We have added the definition of the function 'Transfer' on page 4.
>
> - In Equations (3) and (4) of the updated paper, we have replaced the function 'Classify(M, x)' with 'M(x)' which denotes the output of model M on input x.
>
> - We added the clarification that H is the cross-entropy loss on page 5.
>
> - CW-L2 has been changed to CW-$L_\infty$ in the conclusion.

---

### Official Review · AnonReviewer1 · 2020-10-28
**Novel idea in NLP word with interesting loss**

**Rating:** 7
**Confidence:** 3

**Review:**

##########################################################################
Summary:

The paper considers the fingerprinting of a DNN model via usage of specifically designed adversarial examples.
The authors describe a proposed loss function and results of their experiments

##########################################################################
Reasons for score: I vote for weak accept, because the model fingerprinting is an important topic in the modern DL, and authors propose an interesting idea on how one can generate such fingerprints via adversarial attacks

The authors propose new fingerprinting strategy for an already constructed DL model. They demonstrate, that their method can generate conferrable examples and has a solid performance

#########################################################################
Proposed minor improvements:

Abstract: provider trains a deep neural network and
provides many users access -> remove one word "provide"
everywhere: AUC -> ROC AUC, as there can be other curves with areas under them
 training a DNN is costly because of data preparation (collection, organization, and cleaning) and computational
resources required for training and validating the model -> remove one training; better say "validation of a model"
Formulas 2, 3: not clear what does Classify(S, x) mean
Formula 4: is $\sigma$ a sigmoid function? please, specify
Formula 5: is H a categorical cross entropy? please, specify
Figure 3: what is BIM?

---

> ### Author Response · Authors · 2020-11-20
> **All changes have been incorporated**
>
> We thank the reviewer for their positive feedback and considerations to improve the paper.
> All the reviewer's suggestions have been added to the updated paper.
> The definitions are as follows.
>
> - Classify(M, x) is the classification output of model M on input x. In the updated revision we refer to M(x) instead of Classify(M, x) to improve clarity.
>
> - In Formula 5, $\sigma$ refers to the Softmax function.
>
> - In Formula 6, H refers to the cross-entropy loss.
>
> - In Figure 3, BIM refers to the Basic Iterative Method [1].  We have added a reference to BIM on page 5 in the main part of the paper.
>
> -----------------
>
> [1] Kurakin, Alexey, Ian Goodfellow, and Samy Bengio. "Adversarial examples in the physical world." arXiv preprint arXiv:1607.02533 (2016).

---

### Official Review · AnonReviewer3 · 2020-11-06
**important problem and nice initial results**

**Rating:** 6
**Confidence:** 4

**Review:**

This paper studies fingerprinting a neural network model by using adversarial example techniques. The idea itself is interesting enough, the this work presents a neat development toward solving this problem. An important issue with this problem is to distinguish a reference model from a stolen model. Thus a desire property of the fingerprint adversarial example is to mislead all surrogate models but non reference models. Since adversarial examples are typically transferable to reference models, thus it is important to distinguish a fingerprint from a transferable adversarial example. For a long time, researchers do not have an answer to whether this is possible, and this work provides an evidence that it may generate a conferrable but not transferable adversarial example to achieve the goal.

The idea is pretty simple: we construct a conferrable score function to induce all surrogate models to produce a target label; while all reference models to produce different labels. The experiments show that such an approach can indeed produce some fingerprint adversarial examples to distinguish between reference models and surrogate models to a certain degree.

Having said this, the results are not perfect. For example, Fig 3(b) shows that FGM and PGD have better detectability AUC when epsilon is larger. Although the results are not perfect, this shows promising initial results toward an interesting research domain.

The presentation has some issues that can be fixed by revision. The figures are generally too small to read. Authors can increase them by leveraging one more page. Also, it's necessary to include the definition of FTLL, FTAL, RTLL, RTAL in the main text. Again, one more page should be able to fix the issue.

---

> ### Author Response · Authors · 2020-11-20
> **All changes have been incorporated**
>
> We thank the reviewer for their positive feedback that provided us with the opportunity to improve the clarity of the paper.
>
> We have moved the definition of the model modification attacks 'FTLL', 'FTAL', 'RTLL' and 'RTAL' to the main section of the paper and increased the size of the figures.

---

### Decision · Program_Chairs · 2021-01-07
**Final Decision**

**Decision:**

Accept (Spotlight)

**Comment:**

The paper presents a new idea for detection of model stealing attacks. The new method generates "fingerprint", i.e., adversarial examples that transfer to surrogate models (extracted in model stealing attacks) but not to reference models (i.e., models obtained independently from the same data). If a model owner suspects that some model is stolen, fingerprints can be used for verifications of such claims.

The paper's contribution is novel and significant. It is the first practical tool, to my knowledge, suitable for a reliable characterization of stolen models. The empirical results are quite impressive demonstrating the detection of stolen models with an AUC = 1.0. Some presentations issues have been addressed by the authors during the revision.